# Body image perception and physical activity behavior among adult population: Application of trans-theoretical model of behavior change

Mozhgan Mahdifar[1☯], Seyedeh Belin Tavakoly Sany[2☯], Hadi Tehrani[3,4], Vahid Ghavami[3,5], Mohammad Vahedian Shahroodi[3,4]*

1 Department of Health Education and Health Promotion, Student Research Committee, Mashhad University of Medical Sciences, Mashhad, Iran, 2 Department of Health, Safety, and Environment, School of Health, Mashhad University of Medical Sciences, Mashhad, Iran, 3 Social Determinants of Health Research Center, Mashhad University of Medical Sciences, Mashhad, Iran, 4 Department of Health Education and Health Promotion, School of Health, Mashhad University of Medical Sciences, Mashhad, Iran, 5 Department of Biostatistics, School of Health, Mashhad University of Medical Sciences, Mashhad, Iran

☯ These authors contributed equally to this work.
* vahedianm@mums.ac.ir

**Data Availability Statement:** All relevant data are within the manuscript and its Supporting information files.

## Abstract

### Background

The human body changes during life, but research on how sociodemographic characteristics and physical activity (PA) related to all aspects of body image are still unclear and there is conflict in the relevant results. This study aims to examine gender-specific body image perception and physical activity in relation to BMI and sociodemographic characteristics among health employees who worked in health and medical care centers.

### Methods

The study was a population-based cross-sectional survey of 170 health employees (55 men and 115 women) in Binaloud, Iran. Participants completed a self-administered questionnaire, which consisted of a self-administered Multidimensional Body-Self Image Questionnaire (MBSRQ) and stages of change scales. The full version of Marcus-TTM based on the stages of physical activity changes was used based on 5 items related to regular physical activity behavior and intentions.

### Results

There were no significant differences between men and women in total body image score, body area satisfaction, and attitude. Disease orientation (p = 0.02) and health orientation (p = 0.05) were the only significant differences between men and women. The number of children, level of education, BMI, and PA had a stronger influence on body image concerns and body dissatisfaction. PA and 7 of the 10 subclasses of MBSRQ (appearance evaluation, fitness evaluation, fitness orientation, health evaluation, health orientation, and illness orientation) showed a significant difference (P<0.05) between participants in the five TTM stages.

**Funding:** The author(s) received no specific funding for this work.

**Competing interests:** The authors declare that they have no competing interests.

**Abbreviations:** AE, Appearance evaluation; AO, Appearance orientation; BASS, Body area satisfaction; CCA, Cross-Cultural Adaptation; CVI, content validity index; CVR, content validity ratio; FE, Fitness evaluation; FO, Fitness orientation; HE, Health evaluation; HO, Health orientation; IO, Illness orientation; MBSRQ, self-administered Multidimensional Body-Self Image Questionnaire; PA, PA; SIB, comprehensive network; SW, Self-classified weight; TTM, trans-theoretical model of behavior change; WP, Overweight preoccupation.

## Conclusion

The ideal body image and body satisfaction may differ by the number of children, level of education, gender, BMI, and PA. Evidence from this study supports that participants' stages of change affect their physical activity and body image perception. This result suggests a need for conducting work/home place intervention to promote adults' body image perception and body area satisfaction based on the usage-specific stage under consideration.

## Background

Sedentary behavior and physical inactivity (PA) contribute to managing and preventing non-communicable disease mortality. Individuals with sufficient PA have a 20% to 30% decreased risk of death compared to people with insufficient PA. World Health Organization (WHO) defines PA as "any bodily movement produced by skeletal muscles that requires energy expenditure"[1]. Both moderate- and vigorous-intensity PA include all movement during leisure and work time and for transport to get to and from places. Although several studies have highlighted the significant effect of PA on public health, yet 20% of the general population remains physically inactive [1, 2]. Iran has been among the middle-income countries with medium to high insufficient PA (IPA) in global reports. From 1990 to 2017, the percentage of IPA increased 2-fold in Iran and 1.5 times worldwide, leading to premature mortality worldwide (1.2 million deaths) and in Iran (18,000 deaths) [3]. The prevalence of IPA was high in the total population (54.7%) with a significant difference between the two genders (females: 61.9%; males: 45.3%) and age groups (lowest prevalence (51.6%) in the 18–24-year-old and the highest prevalence (56.1%) in the 35–44-year-old) [3, 4]. Although more national programs have been conducted over the last two decades in Iran to improve clinical characteristics and health status, a large adult population suffers from comorbidities that are associated with IPA [5–7] such as higher BMI, hypertension, history of diabetes, hypercholesterolemia, breast and colon cancers, and cardiovascular disease [7–9]. Therefore, the IPA rate is at an unacceptably high level in Iran and remains a health priority that needs a maintained intervention program because it is still unclear what potential determinate involves an individual's PA behavior and when individuals can change and maintain their PA behavior in a longtime [10–12].

In health and sports psychology research, body image has been regarded as an important factor related to PA and healthy behavior [13, 14]. Body image is a multidimensional construct that reflects attitudinal and perceptual dimensions about one's physical function and appearance. The cognitive dimension evaluates thoughts about one's body function and appearance, whereas the perceptual dimension includes how one describes and sees their body function and appearance [15–17]. The affective dimension consists of the individual's emotion and feelings about their body function and appearance, and the behavioral dimension focus on behaviors resulting from thoughts, perceptions, and feelings about body function and appearance [16, 18]. Body dissatisfaction and body image are gendered phenomena. Previous studies have shown that low levels of body dissatisfaction are common among young people. They found that 22% of men and 24% of women were dissatisfied with their bodies [19, 20]. Likewise, results from these studies suggest that negative attitudes toward body function and appearance, which occur frequently in young men and women, and it is related to increased eating disorders and risk of depression, and reduced quality of life, health, and life satisfaction [19, 21, 22]. Given that body dissatisfaction and body image attitudes are related to the quality of life and human health, it is important to examine modifiable factors associated with positive attitudes

toward body function and appearance and body image disturbance [21, 23]. To date, a significant proportion of clinical studies indicated psychological and medical conditions [24, 25], obesity [24, 26], and sexual orientations [25, 27] may affect body image concern and perception. However, the literature review highlighted the need for further research to examine all aspects of objectively measured and self-reported body image and body dissatisfaction, in the context of the socio-demographic characteristics of different adult groups [1, 21, 32].

The transtheoretical model of behavior change (TTM) is one of the approaches that could help us to explain when and how people are likely to change their PA behaviors [28]. The TTM has been recommended as a coherent theory of change system, and it is suitable for understanding PA behavior through a series of stages. The stages of change include pre-contemplation (unaware that their behavior is producing negative consequences and not thinking about making changes in body image), contemplation (aware that their behavior may be problematic and thinking to start the healthy behavior but not immediately), preparation (planning to take action within the next 30 days and may already be taking small steps to change their body image), action (change their behavior in the last 6 months and intend to continue that behavior change), and maintenance (sustained their behavior for 6 months to change their body image) [36, 37]. A growing number of studies have indicated a significant association between the TTM and different levels of PA and processes of change [28, 29]. However, examining the mediators and moderator of stage transition need further studies.

The relationship between PA behaviors, social-demographic characteristics, and human body image is still unclear, and there are conflicting results [30, 31]. Many of the studies on body image have focused on testing how the attitudinal component of body image was associated with PA and human health [31]. The results of other studies have shown that exercise and PA can cause major changes in weight, appearance, and body shape, and build self-confidence and self-efficacy, all of which can improve body image[13, 19, 32]. To our knowledge, no study has been conducted on examining Iranian health employee's PA behaviors and body image concerns based on the TTM. Likewise, sports medicine professionals and health educators refer to TTM studies as sources and useful guidance to understand the need to encourage people to attend regular PA by promoting their level of processes of change. However, future research that examines cross-sectional differences between the main stage of the TTM are of limited use [28, 33]. Therefore, this study was aimed at applying the TTM to examine gender-specific body image perception and physical activity in relation to BMI and sociodemographic characteristics among health employees who worked in health and medical care centers. In this study, we assume that adult males and females with different sociodemographic characteristics have significantly different body image perception and PA behavior. Further, we expected the 5 stages of the Trans-Theoretical Model (TTM) could be sensitive and reliable to distinguish the participant's decision-making toward increasing their PA behaviors and body image perception from their knowledge levels and values. This understanding will help health educators and professionals design interventions to improve body image.

## Method

### Study design and participants

This study is a population-based cross-sectional survey conducted in Binalod, Iran from July 2020 to September 2020. *The target population included employees working in health care if they met all the following inclusion criteria: (a) were adult population (over 18 years of age); (b) able to give informed consent; (c) did not suffer chronic disease or hospitalized in the past three months; and (d) did not have psychological disorders and sexual orientations. Participants were also excluded if they were pregnant and had not filled out all questionnaire items. To monitor*

*people's eligibility, their medical records were first checked via Electronic Health Records (EHRs). These private and secure lifetime records explain an individual's health history and care. This record system is made up of information from different sources (e.g., doctors, health care clinics, hospitals, laboratories, and pharmacies) that facilitate access to individual medical records, promote the accuracy of treatment decisions and the quality of care, improve clinical research, and reduce medical cost. After that, we interviewed the people and during the interview, we asked them verbally about their medical and mental status, and their weight was also calculated.*

The sample size was calculated using the following formula:

$$n = \frac{Z_{1-\alpha/2}^2 \sigma^2}{d^2}$$

Where, n is the sample size, α is the first type, Z is the table-based normal distribution index that is considered at 5% type 1 error (P<0.05), σ represents the small variable variance, and d shows the accuracy of quantitative variable estimation. According to a previously published study that was conducted to evaluate PA and BMI among employees working in medical institutions in Iran, a first type error, σ, and d equal to 0.05, 4.31, and 0.68, respectively. After adjusting for the non-response of 10%, 170 employees were considered as the sample size.

Since the target population includes all employees working in Binaloud's health care organizations, if meet inclusion criteria. A total of 216 employees worked in Binaloud's healthcare organizations, of which 192 met the inclusion criteria, and 22 did not answer all the items in the questionnaire. Finally, 170 employees (55 men and 115 women) were included in the data analysis and completed all questionnaires. The participation rate of women and men was 92.5% and 60% in this study, respectively. This sample size was sufficient to check the image state of the human body with a 95% confidence level and an accuracy rate of 3.0% for women and 2% for men. The analysis was limited to the adult population (over 18 years of age) interviewed one by one. They also completed a written informed consent form and all questionnaires consisting of self-report items designed to study variables. In each interview, all women and men were asked about their sociodemographic characteristics (age, gender, marital status, income, number of children, and education level) to fill out the questionnaire. In addition, their weight, height, and body mass index (BMI) were measured by two authors (M.M and M. V.SH).

## Measures

**Body image.** Participants completed a questionnaire that included a self-administered Multidimensional Body-Self Relations Questionnaire (MBSRQ) [34] and a trans-theoretical model (TTM) measure [35]. The MBSRQ is a validated self-report measure for the evaluation of all aspects of body image. The full version of MBSRQ includes 69-item and 10 subscales (appearance evaluation and orientation, fitness evaluation and orientation, health evaluation and orientation, illness orientation, the body area's satisfaction scale, the self-classified weight scale, and the overweight preoccupation scale) [34] (Table 1). Each subscale is interpreted according to low and high scores. The questionnaire was translated into Persian according to the International Guidelines for Cross-Cultural Adaptation (CCA) [36], and tested for content validity ratio (CVR), content validity index (CVI) and internal consistency. The CVR, CVI, and overall Cronbach's alpha of the questionnaire were 0.87,0.89, and 0.87, which were acceptable in this study.

**Trans-theoretical model (TTM).** The TTM is a model with a focus on the individual's decision-making, and it assumes that individuals will not change behavior decisively and quickly. Behavior change occurs continuously through five stages in which people try to

**Table 1. Summary of all subclasses of self-reported multidimensional body-self relations questionnaire (MBSRQ).**

| Subclasses | Objectives | Statement |
|---|---|---|
| BSRQ (S: 57–285) Appearance Evaluation (AE), $\alpha$ [b] = .83, $n^a$ = 7, [c] S = 7–35 | Clarify feelings of satisfaction or attractiveness with one's looks. | To select how much, they agree with their appearance: e.g., "most people would consider me good looking or I dislike my physique". From 1 (definitely disagree) to 5 (definitely agree). |
| Appearance Orientation (AO), $\alpha$ = .91, $n$ = 12, S = 12–60 | Extent of investment in one's appearance | To rate how much, they pay attention to their appearance, place more importance on how they look, and engage in much effort to "look good: e.g., I am careful to buy clothes that will make me look my best" or "before going out, I usually spend a lot of time getting ready". From 1 (definitely disagree) to 5 (definitely agree). |
| Fitness Evaluation (FE) $\alpha$ = .81, $n$ = 3, S = 3–15 | Examine feelings of being physically fit or unfit | To rate how much, they feel physically fit, "in shape," or athletically competent: e.g., "I would pass most physical fitness tests" or "my physical endurance is good". From 1 (definitely disagree) to 5 (definitely agree). |
| Fitness Orientation (FO), $\alpha$ = .91, $n$ = 13, S = 13–65 | Extent of investment in being physically fit or athletically competent. | To rate how much, they involve in regularly incorporate exercise activities to maintain or increase their fitness: e.g., "I know a lot about physical fitness", I do not actively do things to keep physically fit". 1 (definitely disagree) to 5 (definitely agree). |
| Health Evaluation (HE), $\alpha$ = .87, $n$ = 6, S = 6–30 | Evaluate feelings of the freedom from physical illness/ or physical health | To rate how much, they feel healthy and their bodies are in good health: "I am a physically healthy person" or "My health is a matter of unexpected ups and downs". From (definitely disagree) to 5 (definitely agree). |
| Health Orientation (HO), $\alpha$ = .89, $n$ = 8, S = 8–40 | Extent of investment in a physically healthy lifestyle. | To clarify how much, they try to lead a healthy lifestyle: e.g., "Good health is one of the most important things in my life." Or "I have deliberately developed a healthy lifestyle." From 1 (definitely disagree) to 5 (definitely agree). |
| Illness Orientation (IO) $\alpha$ = .94, $n$ = 8, S = 8–40 | Extend of reactivity or alertness to being or becoming ill | To rate how much, they alert to personal symptoms of physical illness and are apt to seek medical attention: e.g., "I pay close attention to my body for any signs of illness". From 1 (definitely disagree) to 5 (definitely agree) |
| BASS (S: 9–45) Body Areas Satisfaction Scale, $\alpha$ = .79, $n$ = 9, S = 9–45 | Examine feelings of satisfaction with discrete aspects of one's appearance. | To rate how much, they satisfy with most areas/aspects (i.e., hair, face, muscle tone, weight, height, lower torso, midtorso, and upper torso) and overall appearance) of their body: e.g., "do you satisfy with Upper torso (chest or breasts, shoulders, arms) of your body? From 1 (very dissatisfied) to 5 (very satisfied). |
| Attitude (S: 6–30) Overweight Preoccupation (OWP): $\alpha$ = .89, $n$ = 4, S = 4–20 | Assess a construct reflecting weight, dieting, eating restraint, and fat anxiety. | To select how much, they concerned about being or becoming fat. From 1 (definitely disagree) to 5 (definitely agree) |
| Self-Classified Weight (SCW), $\alpha$ = .89, $n$ = 2, S = 2–10 | Reflects how one perceives and labels one's weight | To rate how much, they perceive their weight: e.g., "From looking at me, most other people would think I am" or "I think I am". From 1 (very underweight) to 4 (very overweight). |

[a] number of question,

[b] Cronbach's alpha;

[c] Minimum and maximum score

change their behavior. Similarly, different behavioral constructs and theories can be used at different stages of the model [37]. *The scale of change stages includes one question with five change stages. The full version of Marcus-TTM based on the stages of physical activity change was examined using a 5-item based on dichotomous scale (no/yes) related to regular physical activity behavior and intentions [29, 38]. Then, we categorized participants into one of the five stages of physical activity behavior change explained earlier. For example, participants select, "I currently do not engage in moderate or vigorous physical activity on a regular basis, and I do not intend to start physical activity in the next 6 months" if they were in the precontemplation stage* (S1 Table,). *The CVR, CVI, and overall Cronbach's alpha of the questionnaire were 0.95, 0.87, and 90, which was acceptable in this study.*

**Body mass index.** Height and weight were measured with an Anthropometer and a Seca 761 scale to the nearest 0.5 cm and 0.5 kg, respectively: with participants wearing lightweight clothing and without shoes. Body mass index (BMI) is measured based on height and weight.

The BMI is explained as weight (kg) divided by the square body height (m$^2$). Participants were categorized into 4 groups based on their BMI: underweight (BMI<18.5 kg/m$^2$), normal persons (18.5 to 24.9 kg/m$^2$), overweight persons (25.0 and 29.9 kg/m$^2$), and obese (BMI $\geq$30.0 kg/m$^2$) [31].

**PA measure.** The PA was examined using the five validated short questions that corresponds to the following questions: I do regular exercise, I am trying to increase my physical strength, it is important for me to participate in PA, I try to be physically active, and I tend to do regular exercise throughout the year. It was rated on a 5-point Likert scale ranging from 1 (completely disagree) to 5 (completely agree). The overall Cronbach's alpha for these questions was 0.84.

## Statistical tests

For this study, a series of descriptive (mean score, standard deviation, and frequency) were conducted to measure sociodemographic characteristics, the level of MBSRQ aspects, BMI, and PA. Bivariate statistics tests ($\chi$2 and ANOVA with Scheffe's post hoc) were used to measure group differences in quantitative variables. We used Spearman correlation analysis to measure the strength of the association of PA and BMI with all aspects of body image and sociodemographic characteristics. Variables with a correlation coefficient of more than 0.1 were included in the multiple logistic regression analysis models. The multiple logistic regression analysis models were employed to examine whether BMI and PA are empirically associated with demographic variables, MBSRQ, and stage of change. This study considered a 95% confidence interval (CI) and p < 0.05 as the significance threshold. Statistical tests were analyzed for all variables using SPSS Statistics 16 (Chicago, Illinois). This study was conducted after the approval and permission of Mashhad University of Medical Sciences Research Committee IR.MUMS.REC.1398.265 and was conducted with consideration of Helsinki Declaration in all phases of the study. Confidential data treatment was guaranteed. Written informed consent was obtained from the participants. Availability of data and materials Data from this study will not be openly available until planned publication outputs have been completed.

## Results

### Participant characteristics and body image

Most of the eligible participants were married, had bachelor's or master's degrees with 2 children, and most of the participants were normal weight or overweight (S2 Table).

The descriptive characteristics of the male and female participants are presented in Table 2. The mean age and PA of female participants in this study were 35.9 ± 8.3 years old and 16.91±3.4, respectively. The mean age and PA in the male participants were 36.84 ± 7.21 years old and 17.98±2.9, respectively. However, normal weight was significantly prevalent among women (55.9%), whereas obesity was significantly prevalent in men (12.7%). The mean scores for the body image subscale were examined based on the MBSRQ in men and women participants. There were no significant differences regarding the total score of body image (P = 0.68), body area satisfaction (p = 0.29), and attitude (p = 0.75) among men and women. Likewise, significant differences between women and men were only observed in 2 of the 10 body image dimensions. Women demonstrated a greater desire for illness orientation (p = 0.02) and health orientation (p = 0.05). However, men showed higher body area satisfaction than women on four items (muscle tone, overall appearance, weight, and height) (Table 2).

**Table 2. Descriptive characteristics of the participants.**

| Variables | | Women (n = 115) | Men (%) | a P-Value |
|---|---|---|---|---|
| **Age, *years M ± SD*** | Range: 22–55 | 35. 9 ± 8.3 | 36.84 ± 7.21 | 0.5 |
| **Education** | Diploma | 19.5 | 38.2 | 0.09 |
| | Higher Diploma | 7.8 | 7.3 | |
| | Bachelor or Master | 40 | 32.7 | |
| | PhD | 20.9 | 10.9 | |
| | Physicians | 12.2 | 10.9 | |
| **Marriage statues** | Single | 25.2 | 16.4 | 0.19 |
| | Married | 74.8 | 83.6 | |
| **Number of child, %** | 0 | 17.2 | 17.4 | 0.47 |
| | 1 | 28.7 | 28.3 | |
| | 2 | 40.2 | 32.6 | |
| | 3 | 12.6 | 15.2 | |
| | More then 3 | 11 | 6.5 | |
| **BMI, %** | Underweight | 0 | 0 | 0.09 |
| | Normal | 55.9 | 43.6 | |
| | Overweight | 39.6 | 43.6 | |
| | Obese | 4.5 | 12.7 | |
| **Stages of TTM, %** | Pre-contemplation | 19.6 | 21.8 | 0.05 |
| | Contemplation | 33.6 | 23.6 | |
| | Preparation | 29.6 | 20 | |
| | Action | 12.2 | 14.5 | |
| | Maintenance | 6.1 | 20 | |
| **BSRQ** | Appearance evaluation | 28.25 ± 4.1 | 27.6 ± 3.63 | 0.15 |
| | Appearance orientation | 52.29 ± 4.66 | 51.74 ± 4.3 | 0.37 |
| | Fitness evaluation | 11.6 ± 2.25 | 11.95 ± 2.03 | 0.1 |
| | Fitness orientation | 47.37 ± 7.3 | 49.23 ± 6.75 | 0.15 |
| | Health evaluation | 21.72 ± 3.27 | 21.2 ± 2.88 | 0.23 |
| | Health orientation | 32.77 ± 5.58 | 31.49 ± 3.98 | 0.046 |
| | Illness orientation | 21.06 ± 3.69 | 20.05 ± 3.12 | 0.02 |
| **BASS** | Body area satisfaction | 35.93 ± 5.81 | 37.05 ± 5.27 | 0.29 |
| **Attitude** | Self-classified weight | 6.83 ± 1.38 | 6.6 ± 1.6 | 0.7 |
| | Overweight reoccupation | 10.6 ± 1.9 | 10.6 ± 2 | 0.8 |
| **Physical activity M ± SD** | General score | 16.91±3.4 | 17.98±2.9 | 0.05 |
| **MBSRQ *M ± SD*** | General score | 256.45 ± 25.3 | 256.4 ± 21.6 | 0.68 |

±: Showing mean score (standard deviation); n: number of eligible participants; a Testing significant change between men and women population; TTM: Trans-Theoretical Model; MBSRQ: Multidimensional Body-Self Relations Questionnaire.

## Applying the trans-theoretical model (TTM)

As shown in S2 Table, 35(20.7%) participants were in the pre-contemplation stage of physical activity, 48 (28.6%) were in the contemplation stage, 42 (24.8%) were in the preparation stage, 23 (13.35%) were in the action stage of body image, and 23 (13.35) were in the maintenance stage. Women demonstrated significantly higher scores on stages 2 (33.6%) and 3 (29.6%), whereas men demonstrated significantly higher scores on stages 2 (23.6%) (Table 3).

Table 3 illustrates how the stages of the TTM are related to BMI, PA, and body image sub-scales. In each stage, the proportion of participants differed based on the criteria defined for moderate or vigorous physical activity. Individuals with no intensity, frequency, and duration criteria

**Table 3. Stages of change of BMI, physical activity and MBSRQ between different genders.**

| Stage of changes | | Pre-contemplation | Contemplation | Preparation | Action | Maintenance | [b] P-value |
|---|---|---|---|---|---|---|---|
| BMI, M ± SD | Men | 27± 4.6 | 26.1±4.9 | 26.7±3.6 | 25.8± 1.7 | 24.81.9 | 0.02 |
| | Women | 24.7±2.7 | 24.4±3.2 | 24.5±.4 | 23.9±5.2 | 23.9±2.4 | 0.05 |
| | [a] P-value | 0.14 | 0.41 | 0.45 | 0.04 | 0.82 | |
| Age, M ± SD | Men | 35±5.2 | 37.3±6.5 | 36.2±7.8 | 39±10.7 | 37±6.8 | 0.022 |
| | Women | 39.2±8.5 | 35.6±8.1 | 35.7±9.3 | 32.0±9.1 | 35±6.9 | 0.011 |
| | P-value | 0.16 | 0.48 | 0.78 | 0.18 | 0.58 | |
| Physical activity (%) | Men | 16.6±1.6 | 17.1±3.7 | 17.9±3.2 | 19.2±2.3 | 19.6±2.5 | 0.024 |
| | Women | 16.6±3.4 | 15.8±3.6 | 16.7±3.3 | 18.8±1.8 | 20.4±1.5 | 0.013 |
| | P-value | 0.84 | 0.28 | 0.39 | 0.38 | 0.52 | |
| MBSRQ, M ± SD | Men | 245±15.9 | 254±26.2 | 254.5±22.4 | 265.5±15.7 | 266.5±19.9 | 0.009 |
| | Women | 252.9±29.1 | 253.1±24.7 | 256.7±23.3 | 269±26.7 | 278±15.2 | 0.011 |
| | P-value | 0.24 | 0.99 | 0.59 | 0.58 | 0.22 | |
| BSRQ, M ± SD | Men | 187.6±13.9 | 198±21.2 | 197.8±19.5 | 204.2±12.5 | 207.2±15.5 | 0.012 |
| | Women | 196.5±23.5 | 196±21.3 | 199±19.9 | 202.8±21.2 | 218.7±1 | 0.015 |
| | P-value | 0.14 | 0.86 | 0.9 | 1 | 0.07 | |
| Appearance evaluation | Men | 26±2.6 | 26.4±3.5 | 27.8±3.7 | 28.6±2.6 | 29.6±4.4 | 0.036 |
| | Women | 28.6±2.9 | 28±4.1 | 28.1±4.5 | 27.7±5.2 | 30.2±3 | 0.028 |
| | P-value | 0.23 | 0.13 | 0.68 | 0.73 | 0.92 | |
| Appearance orientation | Men | 51.2±3.2 | 52.7±3.2 | 50.4±5.9 | 51.8±5.1 | 52.2±4.4 | 0.47 |
| | Women | 53.6±3.6 | 51.8±4.9 | 51.8±4.6 | 51.7±5.4 | 52.6±2.4 | 0.39 |
| | P-value | 0.58 | 0.55 | 0.43 | 0.81 | 0.92 | |
| Fitness evaluation | Men | 11.2±1.5 | 11.8±2.5 | 12±1.7 | 12.1±1.5 | 12.6±2.4 | 0.042 |
| | Women | 10.7±2.6 | 10.7±2 | 11±2.3 | 11.5±2.1 | 13±1 | 0.032 |
| | P-value | 0.7 | 0.15 | 0.28 | 0.62 | 0.81 | |
| Fitness orientation | Men | 44.9±4 | 47.9±8.8 | 50.5±5.1 | 51.7±5 | 52.3±6.9 | 0.005 |
| | Women | 46.1±8.6 | 45.5±7.5 | 47.8±5.1 | 49.1±6.3 | 54.8±3.3 | 0.007 |
| | P-value | 0.44 | 0.42 | 0.15 | 0.39 | 0.58 | |
| Health evaluation | Men | 20.9±3.3 | 20.3±2.7 | 21.1±3.7 | 21.8±1.1 | 22±2.5 | 0.042 |
| | Women | 21.6±3.9 | 21.5±3.3 | 21.1±3 | 22.8±2.4 | 23.7±1.9 | 0.037 |
| | P-value | 0.48 | 0.25 | 0.84 | 0.23 | 0.17 | |
| Health orientation | Men | 28.5±3.5 | 31.9±4.7 | 31.9±2.9 | 32.3±3.9 | 33±3.2 | 0.023 |
| | Women | 32.2±8.4 | 32.1±5 | 32.5±4.8 | 34.2±3.5 | 36.1±2.9 | 0.032 |
| | P-value | 0.08 | 0.84 | 0.47 | 0.29 | 0.09 | |
| Illness orientation | Men | 20±3.8 | 21.1±2.7 | 18.6±3.4 | 20.2±3.3 | 20±2 | 0.056 |
| | Women | 19.5±5.2 | 21.3±3.4 | 21.5±3.1 | 20.8±2.7 | 22.8±1.7 | 0.043 |
| | P-value | 0.98 | 9 | 0.02 | 0.65 | 0.01 | |
| Body area satisfaction | Men | 36±4.8 | 35.5±6.3 | 35.6±4.4 | 40.7±4 | 38.5±4.9 | 0.041 |
| | Women | 34.5±6.6 | 36±5.5 | 36.6±5.4 | 35.2±6.3 | 37.7±5.7 | 0.036 |
| | P-value | 0.49 | 0.68 | 0.38 | 0.37 | 0.85 | |
| Attitude | Men | 17±2.4 | 17.9±2.9 | 18±2.5 | 17.3±3 | 16.6±2.4 | 0.16 |
| | Women | 16.9±2.8 | 17.2±2.1 | 17.5±2.7 | 17.4±2.8 | 17.2±3.1 | 0.32 |
| | P-value | 0.97 | 0.25 | 0.49 | 0.8 | 0.68 | |
| Self-classified weight | Men | 7±1.6 | 6.8±1.5 | 7.18±1.6 | 6.3±0.7 | 6.6±1.02 | 0.13 |
| | Women | 6.4±1.6 | 6.9±1.6 | 6.7±1.6 | 6.2±1.6 | 5.8±1.3 | 0.19 |
| | P-value | 0.1 | 0.69 | 0.38 | 0.32 | 0.37 | |

*(Continued)*

**Table 3.** (Continued)

| Stage of changes | | Pre-contemplation | Contemplation | Preparation | Action | Maintenance | [b] *P-value* |
|---|---|---|---|---|---|---|---|
| **Overweight preoccupation** | Men | 10.1±1.7 | 11.07±1.9 | 10.9±1.8 | 11±2.7 | 10±1.9 | 0.30 |
| | Women | 10.4±2.4 | 10.2±1.6 | 10.1±2 | 11.2±1.8 | 11.4±.1 | 0.34 |
| | P-value | 0.57 | 0.1 | 0.96 | 0.48 | 0.24 | |

±: Showing mean score (standard deviation); n: number of eligible participants;

[a] Testing significant change between men and women population;

[b] Testing significant change between 5 stages of TTM; MBSRQ: Multidimensional Body-Self Relations Questionnaire; BMI: Body Mass Index.

*staged the greatest number of pre-contemplation and contemplation (30%) and the fewest number in maintenance (10.6%) or action (12.9%). The response format of TTM showed small increases in physical activity from pre-contemplation to preparation, while the level of physical activity significantly increased as individuals moved from preparation to the action stage in both genders.* Analyses in Table 3 confirm that 7 of the 10 subclasses of MBSRQ (appearance evaluation, fitness evaluation, fitness orientation, health evaluation, health orientation, and illness orientation), and the general scores of MBSRQ, BSRQ, and BASS showed a significant difference (P<0.05) between participants in the five TTM stages in both genders. The mean score of these subclasses increased across the continuum of the TTM stages from low to high, whereas the mean score of BMI in male participants significantly decreased (P = 0.02) across the continuum of the TTM stages from high to low.

Likewise, Illness orientation (in men), age, attitude (self-classified weight and overweight preoccupation), and appearance orientation (in men and women) showed insignificant difference (P>0.05) between participants in the five TTM stages. Age analysis showed that male participants in the action and maintenance stages (stages 4 and 5) had higher age compared with participants in Pre-contemplation stage (stage 1), whereas women demonstrated significantly higher age on stage 1. Further, the mean score of appearance evaluation and appearance orientation in female participants in the preparation (stage 3) and action (stage 4) stages showed lower scores than the pre-contemplation participants (Stage 1) (Fig 1).

### Association analysis

We tested the correlations between MBSRQ subscale, PA, BMI, and demographic characteristics. Correlation analysis showed that the general score of MBSRQ was significantly correlated (P<0.01) with the TTM stages (r = 0.240), BASS, (r = 0.630), PA (0.583), and BSRQ (0.978). Body area satisfaction was significantly related (p<0.01) with the PA (0.290), BSRQ (0.471), Stage of TTM (r = 0.164), and BMI (r = -0.233), while body image attitude was only correlated with BMI (r = 0.489) and education level (r = 0.179) (S3 Table). According to the results of the correlation analysis, only age and education were significantly related. Therefore, only these variables were included in the study. All variables whose correlation coefficient with PA and BMI was greater than 0.1 were included in the regression analysis. Additionally, multiple logistic regression analysis was used to predict the odds of BMI, and PA, stage from the study variables Tables 4 and 5). Male gender, BASS, and attitude were significantly associated with BMI. Further, TTM, gender, and BSRQ were significantly associated with PA.

### Discussion

Researchers in this study examined the status of all subscales of body image, BMI, and PA across five stages of TTM, and tested associations with BMI, PA, and sociodemographic

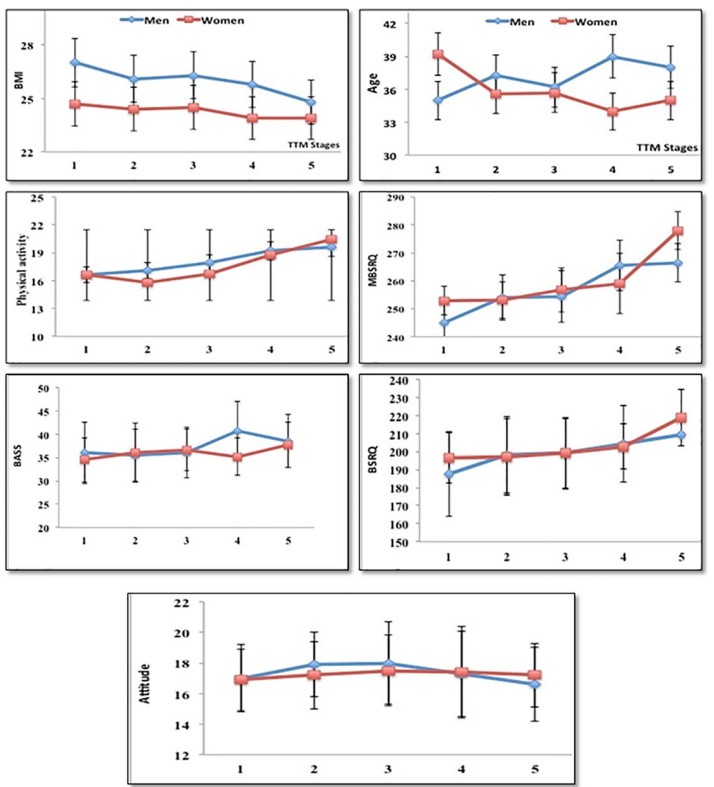

**Fig 1. Stages of change of BMI, PA, age, and MBSRQ between different gender.**

**Table 4. Multiple regression analysis predicting BMI variation from study variables.**

| Predictors | | OR | 95% CI | | P-Value |
|---|---|---|---|---|---|
| | | | Lower | Upper | |
| **Age** | | 1.033 | 0.02 | 1.03 | 0.2 |
| **Gender** | Women | 1 | | | |
| | Men | 4.964 | 1.938 | 12.710 | 0.001 |
| **Education** | Physicians | 1 | | | |
| | Diploma | 2.66 | 0.89 | 2.8 | 0.3 |
| | Higher Diploma | 1.31 | 2.7 | 2.05 | 0.4 |
| | Bachelor or Master | 2.49 | 0.7 | 2.5 | 0.2 |
| | PhD | 1.42 | 1.4 | 2.1 | 0.69 |
| **Marriage statues** | Single | 1 | | | |
| | Married | 0.561 | 0.129 | 2.434 | 0.440 |
| **Number of children** | 2≤ | 1 | | | |
| | 0 | 1.160 | 0.249 | 5.403 | 0.850 |
| | 1 | 0.667 | 0.207 | 2.151 | 0.498 |
| **BASS** | Body area satisfaction | 0.902 | 0.824 | 0.988 | 0.026 |
| **Attitude** | Self-classified weight | 6.8 | 1.1 | 6.9 | 0.000 |
| | Overweight preoccupation | 1.841 | 1.546 | 2.192 | 0.02 |

OR = odds ratio, 95% CI = 95% confidence level, p = probability level

**Table 5. Multiple regression analysis predicting physical activity variation from study variables.**

| Predictors | | OR | 95% CI | | P-Value |
|---|---|---|---|---|---|
| | | | Lower | Upper | |
| **Gender** | Women | 1 | | | |
| | Men | 0.26 | 0.08 | 0.84 | 0.02 |
| **TMM** | 5 stages of trans-theoretical model | 1.019 | 1.004 | 1.035 | 0.013 |
| **BSRQ** | General score of body-self image | 1.652 | 1.002 | 1.726 | 0.036 |
| **BASS** | Body area satisfaction | 1.118 | 0.985 | 1.269 | .084 |
| **Attitude** | Self-classified weight | 0.56 | 0.22 | 1.4 | 0.23 |
| | Overweight preoccupation | 0.88 | 0.26 | 2.9 | 0.83 |

OR = odds ratio, 95% CI = 95% confidence level, p = probability level,

characteristics towards body image concerns in Iranian men and women. These results are important given that few studies so far have specifically focused on BSRQ, BASS, and body image attitude to assess the status of body satisfaction, body image concern, body image attitude, PA, and weight bias across five stages of TTM.

## Characteristics of body image subscales

Based on the results, men's and women's general body image, body satisfaction, and attitude have not differed from each other. We found that body image concerns and body dissatisfaction affect both sexes, particularly about the desire to be thin. The comparison between ideal figures showed that women want a slimmer and thinner body while men want to desire a muscular body that is due to their physical strength, power, and hardness. In this study, body image concerns have become common, and it is considered "normative discontent" in Iranian culture [39, 40]. The choice of a thin and muscular ideal body is the main character in Iran like in other European countries (France, Italy, and Greece), may cause more body image concern and a consequent situation of body dissatisfaction [23, 41, 42]. Likewise, most participants reported more fear about taking a negative appearance and fitness evaluation and a greater attitude toward thinness. Most studies concerning body image have shown that a sizeable number of adult men and women are concerned about their body shape and weight as well [14, 30, 32]. Internalization of the sociocultural environment, including media portrayal, peer and parent influence, cultural invasion, and modern lifestyle is the key psychological process that has reinforced this fear and concern [16, 30, 43].

Our findings provided evidence for a number of gender differences: men and women significantly differ in illness orientation and health orientation, which is in accordance with previous studies. Women were more agreed with illness and health orientation than men on two items, "Good health is one of the most important things in my life" and "I pay close attention to my body for any signs of illness". This indicates that women tried to lead a healthy lifestyle and were more sensitive to personal symptoms of physical illness compared with men. This female behavior could be derived from the social and cultural pressure to be the proper appearance, possibly leading to a negative effect on self-esteem and life quality [36, 37].

As expected, women showed lower body satisfaction compared with men on four items including muscle tone, overall appearance, weight, and height. This result is consistent with previous studies indicating that men are more satisfied with their current body shape status than women [3, 32]. However, several studies showed that women, whether normal weight or overweight, are at greater risk of eating disorders and body dissatisfaction than men [13, 19].

Likewise, it is worth noting that about one-third of the men, participants were not pleased with their physical shape, and they somewhat agreed with the item "I am not as attractive as I would like to be". Our findings for such gender differences have been more diverse. In this study, although good health and signs of illness may be more important to women, men may wish to achieve a more muscular ideal and increase their body size. Male body image and satisfaction are less studied, and it could be different in some aspects of body image than females [13, 40]. Owing to media portrayal focusing on men's body appearance, self-objectification has also affected body image concerns and body dissatisfaction in men [3, 42]. Empirical studies explained that men and women obtain several messages regarding ideal body size, while body satisfaction for men often focuses on body function (power and strength)[44, 45], ideal bodies for women are usually reported to play a role in functioning, sexual behavior, and satisfaction [19, 23, 40]. However, some research findings showed that ideal body size and muscularity for men might be important for sexual satisfaction in women. Longitudinal surveys evaluating gender differences in the role of body satisfaction are still scarce, and it is far from being clearly defined [13, 40].

## Physical activity and BMI association with body image subclasses

In our sample, 50% of participants were overweight or obesity status and had moderate levels of PA, which is lower than the optimal level. However, one-third of the participants did not meet the PA recommendations, and they ignored regular PA in their daily program. BMI was significantly associated with body satisfaction and body image attitudes (self-classified weight and overweight preoccupation) in both women and men. In general, in this study, body satisfaction significantly decreased with increasing BMI. Overweight or obese participants were unhappy with their appearance and the size of their body areas, and they were more concerned about being or becoming fat. Therefore, we expected that body dissatisfaction and negative attitudes toward appearance would be more prevalent among participants with a higher BMI. This result is in line with other studies showing a significant association between body image attitudes and BMI categories[13, 14].

We also explored the significant relationship between PA and body satisfaction and body image concerns. Of note, regression analysis showed that body image concern significantly decreased with increasing PA. It seems that body image concerns and weight management could be high motivators for PA participation. According to the general patterns of results, individuals who participated in regular PA reported higher scores on positive body image perception and body satisfaction. This is generally in agreement with other studies that indicated individuals who participate in PA show higher scores on their body image construct [13, 46]. It seems that people involve PA to modulate their body image concerns, which is very close to what has been shown in Western countries.

Based on our findings, men's PA was more than women's probably because men were more satisfied with their bodies. Higher body dissatisfaction and weight misperception for women might be an important barrier to attending PA and could be engaged in social physique anxiety associated with unreal or real negative physical assessment [3, 47]. Weight misperception among obese and overweight women is linked with less likelihood of attempting or interest in fewer PA [3]. However, it is noteworthy that some published evidence reported the diverse association between PA and body image constructs [16, 34]. It seems that the associations between PA and body image are dynamic, complex, and bidirectional [13] because various aspects might have specific effects on PA and body image concerns such as eating disorders, psychological and medical conditions, bariatric surgery, and sexual orientations. This issue could be considered as new research questions of focus for future study.

Furthermore, the level of PA significantly differed based on the 5 stages of change models. Therefore, we expected participants who were in the maintenance and action stages to perceive greater benefits from PA compared to participants who were in the contemplation and pre-contemplation stages.

### Trans-theoretical model of behavior change

*Consistent with predictions of the TTM, physical activity in both genders is only different between the preparation and action stages because individuals are not engaging in physical activity in pre- contemplation, contemplation, and preparation stages (inactive stages) compared with the action and maintenance stages. In this study, a significant effect was observed from preparation to action points, the stages at which participants are engaging in regular PA. This result is in line with the same study that reported participants engaging in regular activity in the action and maintenance stages, and the difference is only related to times of physical activity between the stages*[29, 33]. *Surprisingly, our finding showed small increases in physical activity from pre-contemplation to preparation, suggesting that transitions could be correlated with changes in physical activity even among inactive stages.*

The results of the present study showed that participants who were in the maintenance and action stages showed greater body image perception, body area satisfaction, and they perceive greater benefit from PA, compared to participants who were in the contemplation and pre-contemplation stage. In addition, male participants who were in the results of the present study showed that participants who were in the maintenance and action stages showed greater body image perception and body area satisfaction, and they perceived greater benefit from PA, compared to participants who were in the contemplation and pre-contemplation stage. In addition, male participants who were in the maintenance and action stage showed lower BMI than participants who were in the contemplation and pre-contemplation. This finding suggested that those who intend to change their behavioral perspectives on body image or have sustained behavior to change their body image might be more successful in improving their body image. Based on more studies, the TTM helped in identifying which participants could be receptive to using body image information that might be implemented to direct the design of educational interventions that may facilitate the implementation of this information at the population level [48, 49].

In this study, an interesting result was also found for age analysis. It showed that male participants in the action and maintenance stages (stages 4 and 5) had higher age compared with participants in the Pre-contemplation stage (stage 1), whereas women demonstrated significantly higher age in stage 1. The results for the number of disorders showed that the mean score of appearance evaluation and appearance orientation in female participants in the preparation (stage 3) and action (stage 4) stages who plan to use behavior to change their body image or irregularly use these behaviors showed lower scores than the pre-contemplation participants (stage 1) who had not thought about it. This indicated that participants in stage 1 found the appearance evaluation and appearance orientation as important and interesting more than those (stages 3 and 4) who were using it irregularly. This result confirms an indicating a mild rejection of body image evaluation as expected.

Although TTM is common in health behavior change research, it has rarely been included in body image perception [48, 49]. In this study, the performance of the TTM was interesting. This model not only provides the pattern for the stage of change of key variables but also may also point to interesting results in research and practice. However, our finding did not decisively show that the model stages sensitively and reliability discriminate between participants because some variables in this study were not supported by the TTM, which is consistent with

evidence from other health studies that were being conducted simultaneously [19, 49]. It seems that this simple dichotomy might be enough to distinguish between different approaches and feelings, but this would need to be tested empirically [13, 48]. Evidence from this study shows that the TTM presents one of the most important efforts to distinguish the participant's decision-making toward improving their body image from their knowledge levels and values. We, therefore, conclude that the TTM is applicable to operationalize different strategies and adult populations.

## Limitations

The design of this investigation was cross-sectional, and it was not possible to examine the temporal criteria of causality and cause-and-effect relationships. It is also worth noting that the use of a self-reported instrument may lead to overestimating or underestimating body image scores, which may result in an increased risk of body image disturbance. Additionally, these results may not be generalizable to the population living outside Iran, the elderly, and adolescents. We recommended future studies on the examination of body image constructs among more diverse populations with different age, BMI, and PA patterns. Finally, since psychological and medical conditions have specific effects on the level of PA and body image perception [24, 27]; therefore, these aspects could be considered as new research questions of focus for future study.

## Conclusion

Given the main findings from this research, PA behavior and body image perception is quite a sensitive issue in both men and women population. Body mass index was significantly associated with body satisfaction and body image attitudes in both women and men. Participants with high BMI were unhappy with their appearance and the size of their body areas, and they were more concerned about being or becoming fat. The level of PA significantly differed based on gender, and body image perception, and the 5 stages of change models. Of note, individuals who participated in regular PA reported higher scores on positive body image perception and body satisfaction. Evidence from this study shows that the TTM presents one of the most important efforts to distinguish participants' decision-making toward changing their PA, and it could be applicable to operationalize different strategies and the adult population. Likewise, a significant change was observed from preparation to action points, the stages at which participants are engaging in regular PA. Therefore, we expected participants who were in the maintenance and action stages to perceive greater benefits from PA compared to participants who were in the contemplation and pre-contemplation stages. We recommended work/homeplace interventions such as media literacy and life skills education to improve individuals' decision-making toward improving their body image. In addition, there is a need for further study to advance theoretical and conceptual strategies to understand the potential associations between body image constructs, PA, PA, and sociocultural influences in different populations.

## Supporting information

**S1 Table. The full version of Marcus-TTM based on the stages of physical activity change.**
(DOCX)

**S2 Table. Subject general characteristics.**
(DOCX)

**S3 Table. Spearman correlations between MBSRQ subscale, PA, BMI, and demographic characteristics.**
(DOC)

## Acknowledgments

The author wishes to express her gratitude towards the vice president of research in Mashhad University of Medical Sciences.

## Author Contributions

**Conceptualization:** Mohammad Vahedian Shahroodi.

**Data curation:** Mozhgan Mahdifar.

**Formal analysis:** Vahid Ghavami.

**Methodology:** Hadi Tehrani.

**Writing – original draft:** Seyedeh Belin Tavakoly Sany, Hadi Tehrani.

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
