## [Decision Letter · Decision Letter 0]

18 May 2023

PONE-D-23-00038Body Image Perception, Weight Concerns and Physical Activity among Adult Population: Application of Trans-Theoretical ModelPLOS ONE

Dear Dr. Vahedian Shahroodi,

Thank you for submitting your manuscript to PLOS ONE. After careful consideration, we feel that it has merit but does not fully meet PLOS ONE’s publication criteria as it currently stands. Therefore, we invite you to submit a revised version of the manuscript that addresses the points raised during the review process. For this manuscript to be accepted for publication, please address all the reviewers comments, in particular, please be more precise and concise in your description of the Transtheoretical Model and your limit the one construct of stages of change.

We look forward to receiving your revised manuscript.

Kind regards,

Jeffrey S. Hallam, Ph.D., FRSPH

Academic Editor

PLOS ONE

Journal Requirements:

5. We note that Figures S1 and S3 in your submission contain map images which may be copyrighted. All PLOS content is published under the Creative Commons Attribution License (CC BY 4.0), which means that the manuscript, images, and Supporting Information files will be freely available online, and any third party is permitted to access, download, copy, distribute, and use these materials in any way, even commercially, with proper attribution. For these reasons, we cannot publish previously copyrighted maps or satellite images created using proprietary data, such as Google software (Google Maps, Street View, and Earth). For more information, see our copyright guidelines: http://journals.plos.org/plosone/s/licenses-and-copyright.

 a. You may seek permission from the original copyright holder of Figures S1 and S2 to publish the content specifically under the CC BY 4.0 license. 

6. Please upload a copy of Supporting Information Table S1 which you refer to in your text on page 13.

Additional Editor Comments:

What are the inclusion criteria described on page 9? The population sample was all employees working in Binaloud's health care organizations (how many employees)?

What instrument was used to determine stage of change? Citation #34 is incomplete on the reference page, however, it does not provide a stages of change instrument for physical activity.

The description of the TTM is incomplete at best. Behavior change, as measured by the stages of change, is only different between preparation and action. There is not physical activity in precomtemplation contemplation or preparation. In the action and maintenance stages, participants are engaging in regularly activity and the only difference between the stages is how long (6-months) participants have been engaging in physical activity. The authors need to describe that they only focused on one variable of the TTM (stages of change).

Stages of changes is a categorical variable (mutally exclusive as participants by definition may be in only one stage) so it should not be analyzed using spearman correlations. If the authors believe (and can support theoretically) that the stages are rank ordered than they should make the case. Marital status is the same either yes or no? is there a rank order? Based on the title of table 4, it is unclear what correlation is being reported. Are they all spearman rank order correlations? Number of children?

Reviewers' comments:

Reviewer's Responses to Questions

**Comments to the Author**

1. Is the manuscript technically sound, and do the data support the conclusions?

Reviewer #1: Partly

Reviewer #2: Yes

2. Has the statistical analysis been performed appropriately and rigorously? 

Reviewer #1: No

Reviewer #2: Yes

3. Have the authors made all data underlying the findings in their manuscript fully available?

Reviewer #1: No

Reviewer #2: No

4. Is the manuscript presented in an intelligible fashion and written in standard English?

Reviewer #1: No

Reviewer #2: Yes

5. Review Comments to the Author

Reviewer #1: Body Image Perception, Weight Concerns and Physical Activity among Adult

Population: Application of Trans-Theoretical Model

The study on body image, BMI, and PA investigated across five stages of TTM, and tested associations with BMI, PA, and socio-demographic characteristics towards body image concerns in Iranian men and women. In spite of sound results found in figure and tables, the methodology, results, and conclusion were not written well and strong. I recommend the authors to revise the manuscript based on major comments.

Abstract

-Methods: How TTM model was implemented in this study? (add these to the methods in abstract and main file).

-Results: which sub subclasses of MBSRQ showed a significant difference should be added to the results.

-Conclusion doesn’t show the main findings of the study.

Background: this part is so long and unrelated to the study objectives. I couldn’t find any related literature review to show the gap of knowledge for the study. Concepts and structure based study related to TTM, physical activity, should be clearly added to the background. This part was not satisfying and written in weak pattern. Figures s 1 and s 2 is not related to this study and only shows the prevalence of outcomes, and the prevalence is not a known parameter for your study. I recommended to remove both supplemental figures from the background. The number of questions used for TTM and samples should be added to the method.

Method:

-Sib is not used in hospitals, It is the Integrated Health Record System, locally known as the "SIB," and is the most used information system for recording public health services and in fact not hospitals.

-in the first part of the method authors declared that all men (n= 92) and women (n= 124) working in Binaloud's health care. And in the next paragraph stated that “the target population includes all employees working in Binaloud's health care organizations, we used convenience sample to select eligible participants easy to contact and met inclusion criteria”.

Results:

-Second line of the results after age, PA result need suitable unite of measurements. Please, add to the section as well as for the male.

- Tbale 2, the name of TTM model with each stage should be written.

-how PA was measured? Which questions were used for the measurements? Please, explain it clearly.

Reviewer #2: The manuscript reports a study about the modification of body image and physical activity across ages in Iranian population. The paper presents specific interesting aspects like the evaluation of body image attitudes in understudied population. The results showed that body image issues are similar to Western countries. However, I think the authors should revised their paper looking at specific aspects that are missing in their paper:

- the introduction is quite long and not focused on the paper's goal. Please revised this aspect.

- have you evaluated psychological and medical conditions? because these aspects might have a role in body image issues. For example, eating disorders have specific effects on body image evaluation and perception, as well as bariatric surgery or extreme obesity, see https://doi.org/10.1002/erv.2812 and https://doi.org/10.1007/s11695-020-05166-z

- have you evaluated gender or sex? please be consistent with this through the manuscript

- have you evaluated sexual orientations? Because this aspect has been shown to be crucial for body image concerns, especially in the male/female comparisons, see https://doi.org/10.1007/s40519-020-01047-7

- is it possible that physical activity and body image concerns are linked in a compulsory way like it happens in eating disorders (see https://doi.org/10.1016/j.eatbeh.2022.101675)? The idea that people use physical activities to modulate their body image concerns is very closed to what has been shown in Western countries and it is very interesting that also your study showed that.

- you reported only obesity as problem related to body, but recent studies have showed the presence also of high rates of eating disorders in the Middle East Countries. Has this aspect been considered?

6. PLOS authors have the option to publish the peer review history of their article (what does this mean?). If published, this will include your full peer review and any attached files.

Reviewer #1: No

Reviewer #2: No

---

## [Author Response · Author response to Decision Letter 0]

18 Oct 2023

Dear editors and reviewers:

Thank you so much for arranging a timely review of our manuscript. We are excited to receive the decision letter from your editorial office. We would like to thank all the members of the editorial team of Journal and the peer reviewers for their great suggestions and remarks. We thank you for the chance to submit a revision. As soon as we received the decision letter, we held a group meeting to address all of the critiques mentioned, with a particular focus on the issues that need to be improved.

To the best of our knowledge, we considered all the topics that required further attention. Some parts of the manuscript were rewritten. We are confident that the present version of the manuscript is much stronger and more clear. We wish to refer to the comments systematically. The changes in the revised manuscript are marked and highlight.

---

## [Decision Letter · Decision Letter 1]

12 Jan 2024

Body Image Perception and Physical Activity Behavior among Adult Population: Application of Trans-Theoretical Model of behavior change

PONE-D-23-00038R1<o:p></o:p>

Dear Dr. Shahroodi<o:p></o:p>

We’re pleased to inform you that your manuscript has been judged scientifically suitable for publication and will be formally accepted for publication once it meets all outstanding technical requirements.<o:p></o:p>

Within one week, you will receive an e-mail detailing the required amendments. When these have been addressed, you will receive a formal acceptance letter and your manuscript will be scheduled for publication.<o:p></o:p>

An invoice for payment will follow shortly after the formal acceptance. To ensure an efficient process, please log into Editorial Manager at http://www.editorialmanager.com/pone/, click the 'Update My Information' link at the top of the page, and double check that your user information is up-to-date. If you have any billing related questions, please contact our Author Billing department directly at authorbilling@plos.org.<o:p></o:p>

If your institution or institutions have a press office, please notify them about your upcoming paper to help maximize its impact. If they’ll be preparing press materials, please inform our press team as soon as possible -- no later than 48 hours after receiving the formal acceptance. Your manuscript will remain under strict press embargo until 2 pm Eastern Time on the date of publication. For more information, please contact onepress@plos.org.<o:p></o:p>

Kind regards,<o:p></o:p>

Roghieh Nooripour, Ph.D

Academic Editor

PLOS ONE<o:p></o:p>

---

## [Editor Report · Acceptance letter]

15 Feb 2024

PONE-D-23-00038R1 

PLOS ONE

Dear Dr. Vahedian Shahroodi, 

I'm pleased to inform you that your manuscript has been deemed suitable for publication in PLOS ONE. Congratulations! Your manuscript is now being handed over to our production team.

Kind regards, 

on behalf of

Dr. Roghieh Nooripour 

Academic Editor

PLOS ONE